# *Vitis vinifera* L. Leaf Extract Inhibits In Vitro Mediators of Inflammation and Oxidative Stress Involved in Inflammatory-Based Skin Diseases

**DOI:** 10.3390/antiox8050134

**Published:** 2019-05-16

**Authors:** Enrico Sangiovanni, Chiara Di Lorenzo, Stefano Piazza, Yuri Manzoni, Cecilia Brunelli, Marco Fumagalli, Andrea Magnavacca, Giulia Martinelli, Francesca Colombo, Antonella Casiraghi, Gloria Melzi, Laura Marabini, Patrizia Restani, Mario Dell’Agli

**Affiliations:** 1Department of Pharmacological and Biomolecular Sciences, Università degli Studi di Milano, 20133 Milan, Italy; chiara.dilorenzo@unimi.it (C.D.L.); stefano.piazza@unimi.it (S.P.); yurimanzoni@alice.it (Y.M.); c.brunelli@biolifeitaliana.it (C.B.); marco.fumagalli3@unimi.it (M.F.); andrea.magnavacca@unimi.it (A.M.); giulia.martinelli@unimi.it (G.M.); francesca.colombo1@unimi.it (F.C.); gloria.melzi1@studenti.unimi.it (G.M.); patrizia.restani@unimi.it (P.R.); 2Department of Pharmaceutical Sciences, Università degli Studi di Milano, 20133 Milan, Italy; antonella.casiraghi@unimi.it; 3Department Environmental Science and Policy, Università degli Studi di Milano, 20133 Milan, Italy; laura.marabini@unimi.it

**Keywords:** *Vitis vinifera* L., grapevine leaves, keratinocytes, skin inflammation, oxidative stress, in vitro skin permeability, TNF-α, UVB, LPS, H_2_O_2_

## Abstract

Psoriasis is a chronic cutaneous condition characterized by the release of pro-inflammatory mediators and oxidative stress. The reduction of these factors is currently the most effective strategy to inhibit the symptoms of pathology. Antioxidants from natural sources are increasingly used to improve skin conditions. Dried red leaves from grapevine (*Vitis vinifera* L., cv Teinturiers) showed anti-inflammatory and anti-bacterial activities, but their possible effects on keratinocytes have not been previously investigated. In this study we tested the ability of a water extract from grapevine leaves (VVWE) to inhibit inflammatory conditions in human keratinocytes (HaCaT cells), challenged with proinflammatory (tumor necrosis factor-α (TNF-α) or lipopolysaccharide (LPS)) or prooxidant (ultraviolet B radiation (UVB) or H_2_O_2_) mediators. VVWE inhibited interleukin-8 (IL-8) secretion induced by proinflammatory stimuli, acting on the IL-8 promoter activity, but the effect was lower when prooxidant mediators were used. The effect was partly explained by the reduction of nuclear factor-κB (NF-κB)-driven transcription and nuclear translocation. Furthermore, vascular endothelial growth factor (VEGF) secretion, a regulator of angiogenesis, was inhibited by VVWE, but not matrix metalloproteinase-9 (MMP-9), a protease involved in matrix remodeling. VVWE, assayed on Franz diffusion cell system, showed a marked reduction of High Performance Liquid Chromatography (HPLC)-identified compounds. Pure molecules individually failed to reduce TNF-α-induced IL-8 release, suggesting synergistic effects or the presence of other bioactive compounds still unknown.

## 1. Introduction

The epidermis, the outermost part of the skin, is the first barrier between our organism and the environment, and this layer is mainly constituted by keratinocytes [1]. This cell population possesses an active role in skin’s defense but is also relevant to the pathogenesis of chronic inflammatory skin diseases, such as psoriasis and atopic dermatitis [2].

Psoriasis is a chronic skin disease affecting approximately the 2% of the worldwide population [3] and is characterized by inflammation, increased dermal angiogenesis, and hyperproliferation of keratinocytes. The pathology has a complex genetic inheritance [4] which causes dysregulation of the innate immune system [5], while an interplay between environmental and genetic factors is responsible for the disease-starting events, such as the release of pro-inflammatory mediators and oxidative stress [6]. Tumor necrosis factor-α (TNF-α) plays a central role in the complex cytokine network of psoriasis, as demonstrated by the clinical efficacy of monoclonal antibody therapy (anti-TNF-α) in psoriatic patients (i.e., with infliximab) [7]. In human keratinocytes, TNF-α induces the activation of pro-inflammatory mediators, including nuclear factor-κB (NF-κB) [8], which translocates from the cytoplasm into the nucleus. Ultraviolet B radiation (UVB) radiations activate the canonical NF-κB pathway [9] through the generation of reactive oxygen species (ROS), which in turn exacerbate oxidative stress [10]. Moreover, NF-κB regulates the expression of several genes involved in skin inflammatory conditions, such as interleukin-8 (IL-8) and vascular endothelial growth factor (VEGF) in different cell types, including keratinocytes [11]. IL-8 is a potent chemokine involved in the recruitment of leukocytes [12] whereas VEGF, which is increased in psoriatic lesions [13], promotes the formation of the typical psoriatic microvasculature [14].

Inhibition of VEGF has shown promising results improving symptoms of psoriasis [15], also in animal models [16]. Furthermore, TNF-α regulates extracellular matrix remodeling through matrix metalloproteinase (MMP) production in keratinocytes [17] including MMP-9 [18]. MMPs are deeply involved in cell migration, tissue remodeling, vasodilatation, and angiogenesis.

Monoclonal antibodies against TNF-α (i.e., infliximab, adalimumab, and golimumab) and circulating receptor fusion proteins (etanercept) are effective therapies for psoriasis, but the search for new strategies is mandatory due to the possible occurrence of serious side effects including lymphoma, infections, congestive heart failure, demyelinating disease, a lupus-like syndrome, induction of auto-antibodies, injection site reactions, and systemic adverse effects [19].

The reduction of cytokines and growth factors is currently the most effective strategy to inhibit symptoms of psoriasis, and different keratinocyte-based assays, using monolayer cultures, are widely employed to assess the effects of pharmacological treatments on proliferation and inflammation [20].

In this contest, antioxidants from natural source, including botanicals, are increasingly used to improve skin inflammatory and oxidative stress conditions [21]. In the last few years, a variety of botanicals have been tested for their antioxidant and anti-inflammatory activities, mostly due to the high content of polyphenols [22,23,24,25]. However, the support for the topical use in psoriasis is limited by the number of studies available in the literature [26].

Grapevine (*Vitis vinifera* L.) is a plant belonging to the genus of Vitaceae, originating in the Mediterranean area. Dried red leaves from the cultivar Teinturiers should contain at least 4.0 percent anthocyanins, expressed as cyanidin-3-*O*-glucoside according to the European Scientific Cooperative on Phytotherapy (ESCOP) Monograph [27]. Grapevine leaves contain a variety of phytoconstituents showing high antioxidant activity including condensed tannins, phenols, and anthocyanins. Moreover, grapevine leaves show other biological properties including anti-inflammatory, anti-bacterial, and vasorelaxant effects [28,29]. We have previously demonstrated that the water extract from *Vitis vinifera* leaves (VVWE) shows anti-inflammatory activity at the gastrointestinal level, acting on the NF-κB pathway [30]. Grapevine leaves are efficiently used in the treatment of varicose veins [31] as formulations for oral or topical use. However, their possible anti-inflammatory effects in keratinocytes have not been previously investigated.

The aim of the present study was to investigate the ability elicited by a water extract from grapevine leaves to inhibit inflammatory conditions induced by mediators of inflammation or oxidative stress in human keratinocytes. To reach this goal, cultured human keratinocytes (HaCaT) cells were used as reliable model of human keratinocytes, and the effect of VVWE on several markers of skin diseases was evaluated following activation with proinflammatory (TNF-α or lipopolysaccharide (LPS)) or prooxidant (UVB or H_2_O_2_) mediators.

## 2. Materials and Methods

### 2.1. Materials

HaCaT cells were purchased from Cell Line Service GmbH (Eppelheim, Germany). Dulbecco’s Modified Eagle Medium (DMEM), 3-(4,5-dimethylthiazol-2-yl)-2,5-diphenyltetrazolium bromide (MTT), and 3,3′,5,5′-tetramethylbenzidine (TMB) were from Sigma Aldrich (Milan, Italy). Penicillin, streptomycin, l-glutamine, sodium pyruvate, trypsin-EDTA, and Lipofectamine^®^ 2000 were from Life Technologies Italia (Monza, Italy).

Human TNF-α, the Human VEGF Elisa Development Kit, and the Human IL-8 Elisa Development Kit were from Peprotech Inc. (London, UK). Fetal bovine serum (FBS), and disposable materials for cell culture were purchased by Euroclone (Euroclone S.p.A., Pero-Milan, Italy). The plasmid NF-κB-LUC containing the luciferase gene under the control of three κB sites was a gift of Nikolaus Marx (Department of Internal Medicine-Cardiology, University of Ulm, Ulm, Germany). The native IL-8-LUC promoter was kindly provided by Takaaki Shimohata (Department of Preventive Environment and Nutrition, University of Tokushima Graduate School, Tokushima, Japan). The promoter contains sequences responsive to several transcription factors such as activator protein 1 (AP-1), CCAAT-enhancer-binding protein-β (C/EBPβ), and NF-κB. Britelite^TM^ plus was from Perkin Elmer (Monza, Italy).

### 2.2. Plant Material and Preparation of the Water Extract (VVWE)

Dried and powdered leaves from *Vitis vinifera* L. cv. Teinturiers were kindly donated by PhytoLab Company (Vestenbergsgreuth, Germany). Plant material was maintained at 4 °C until extraction. VVWE was prepared according to ESCOP monographs [27]. The plant material was extracted twice, at room temperature, in the dark, with deionized water in the ratio herb/water 10 g/100 mL, and lyophilized. The recovery (*w*/*w*) was 26% calculated on the dried drug weight. Samples were then stored at −20 °C until the assays. Before biological evaluations, the extract was dissolved in sterilized distilled water at a concentration of 50 mg/mL, and immediately stored in aliquots at −20 °C.

### 2.3. Cell Culture

HaCaT cells were grown at 37 °C in DMEM supplemented with 100 units penicillin per mL, 100 mg streptomycin per mL, 2 mM l-glutamine, and 10% heat-inactivated FBS, under a humidified atmosphere containing 5% CO_2_.

### 2.4. Characterization of Grapevine Extract by High Performance Liquid Chromatography (HPLC)

VVWE was characterized by a validated HPLC-DAD method as previously described [26]. Briefly, two different HPLC methods coupled with a Diode Array Detector (DAD) have been used, the first for anthocyanin detection at 520 nm by the use of a Synergi 4u MAX-RP 80 Å column (250 × 4.60 mm × 4 μm) (Phenomenex, Torrance, CA, USA) and the second for flavonols and caffeic acid derivative detection at 360 nm by Synergi 4u MAX-RP 80 Å column (250 × 2 mm × 4 μm) (Phenomenex, Torrance, CA, USA). The anthocyanins standards, hyperoside and kaempferol-3-*O*-glucoside, were from Extrasynthese (Genay, France); all the other standards and solvents were bought from Sigma-Aldrich (St. Louis, MO, USA).

### 2.5. IL-8 Release

Cells were grown in 24-well plates for 48 h (30,000 cells per well) before the cytokine treatment. IL-8 was quantified using a Human Interleukin-8 ELISA Development Kit as previously described [30,32]. Preliminary time-course experiments were performed to set the best conditions for further experiments. HaCaT were treated with TNF-α (10 ng/mL), LPS (5 μg/mL), or H_2_O_2_ (100 μM), for 3, 6, 24, and 30 h. The IL-8 secretion induced by TNF-α or LPS was tested after 6 h of treatment, while the 24-h time point was chosen for H_2_O_2_. Epigallocatechin-3-gallate (EGCG) (20 μM) was used as the reference inhibitor of IL-8 secretion.

### 2.6. Transient Transfection Assays

HaCaT cells were plated in 24-well plates and transiently transfected with the plasmid NF-κB-LUC or native IL-8-LUC, both at 250 ng per well, using Lipofectamine^®^ 2000, according to previous studies [32]. Sixteen hours later, the cells were treated for 6 h with increasing concentrations of VVWE in the presence of the pro-inflammatory mediators (TNF-α at 10 ng/mL or LPS at 5 μg/mL). After six hours, cells were harvested and the luciferase assay was performed using the BriteliteTM Plus reagent (PerkinElmer Inc., Massachusetts, USA) according to the manufacturer’s instructions. Data were expressed considering 100% of the luciferase activity related to the cytokine-induced promoter activity.

### 2.7. NF-κB Nuclear Translocation

For the evaluation of the NF-κB (p65) translocation, HaCaT were plated at the concentration of 3 × 10^6^ cells/mL in 100-mm plates with fresh complete medium. After 48 h, cells were treated for 1 h with the inflammatory mediators (TNF-α at 10 ng/mL or LPS at 5 μg/mL) and the extract (10–50 μg/mL) under study, using FBS-free medium. Nuclear extracts were prepared using the Nuclear Extraction Kit from the Cayman Chemical Company (Ann Arbor, MI, USA) as previously described [32].

Data were expressed considering 100% of the absorbance related to the cytokine-induced NF-κB nuclear translocation. EGCG (20 μM) was used as the reference inhibitor of the NF-κB nuclear translocation. The results are the mean ± SD of three experiments in triplicate.

### 2.8. UVB Radiation

HaCaT cells were grown in 24-well plates for 48 h (30,000 cells per well), washed with phosphate-buffered saline (PBS), and exposed to UVB (40 mJ/cm^2^) light source (Triwood 31/36, W36, V230, Helios Italquartz, Milano, Italy) in a glass bath. Radiation time (about 50 s) was adjusted for each experimental day, measuring energy emission by the LP 471 UVB probe (Delta OHM, Padova, Italy). After radiation, serum free fresh culture medium was immediately added. For the evaluation of IL-8 release, cells were treated for additional 9 h, a time point selected in preliminary time-course experiments. For NF-κB (p65) translocation, HaCaT cells were treated for 1 h after UVB radiation. EGCG (20 μM) was used as reference inhibitor.

### 2.9. VEGF Release

Cells were grown in 24-well plates for 48 h (30,000 cells per well) before challenge with the pro-inflammatory mediator. VEGF was quantified by the Human VEGF ELISA Development Kit. Briefly, Corning 96-well EIA/RIA plates (Sigma-Aldrich, Milan, Italy) were coated with the antibody provided in the ELISA Kit (PeproTech Inc., London, UK) overnight at 4 °C. After blocking the reaction, 300 μL of samples were transferred into wells at room temperature for 2 h. The amount of VEGF in the samples was detected by spectroscopy (signal read: 450 nm, 0.1 s) as described above. The quantitative measurement of VEGF was done using an optimized standard curve supplied with the ELISA set (100–2000 pg/mL). The maximal release of VEGF was observed at 24 h for TNF-α (10 ng/mL) and H_2_O_2_ (100 μM) or 30 h for LPS (5 μg/mL). EGCG (20 μM) was used as the reference inhibitor of VEGF secretion.

### 2.10. MMP-9 Release

MMP-9 secretion was performed on HaCaT cells treated with TNF-α (10 ng/mL) and (LPS 5 μg/mL) for 3, 6, 24, and 48 h. Cells were grown in 24-well plates (30,000 cell/well) for 48 h, before the treatment. Human MMP-9 ELISA Kit from RayBio^®^ (Norcross GA, USA) was used to quantify MMP-9 secretion according to manufacturer’s instructions. 300 μL of samples in duplicate were transferred into a 96-well plate coated with anti-human MMP-9 and incubated overnight at 4 °C with gentle shaking. MMP-9 secreted was detected by the use of biotinylated and streptavidin–horseradish peroxidase (HRP) conjugate antibodies as previously described. The quantitative measurement of MMP-9 was done using an optimized standard curve supplied with the ELISA kit (8.23–6000 pg/mL). The MMP-9 secretion reached the maximum at 24 h for both TNF-α and LPS. VVWE was tested at 10–50 μg/mL in the presence of stimuli and between 2.5–200 μg/mL without the pro-inflammatory mediators.

### 2.11. Cytotoxicity Assay

The cell morphology before and after treatment was assessed by light microscope inspection. Cell viability was assessed by the MTT test and verified by lactate dehydrogenase (LDH) assay. No sign of cytotoxicity was observed in cells treated with VVWE at 5–500 μg/mL for 6 h.

### 2.12. In Vitro Skin Permeation Study

The human epidermis membrane used for in vitro permeation studies was obtained from the abdominal skin of a single donor. The epidermis was prepared according to the heat separation method, as previously reported [33].

Ex vivo skin permeation study: the study was performed using the Franz diffusion cell method. The human epidermis was mounted carefully on the receiver compartment of the Franz’s cell with the stratum corneum side in contact with donor solution. The receiver compartment was filled with freshly prepared degassed HCl 0.1 M solution (receiver phase). VVWE (150 mg) solubilized in 0.1 M HCl:EtOH (50:50, *v*/*v*) was loaded in the donor compartment (0.5 mL). At predetermined intervals (1, 5, 7, 24, 32, 48 h), 0.2-mL samples were removed from the receiver compartment and immediately replaced with fresh receiver phase. Sink conditions were maintained throughout the experiment. The samples were assayed by HPLC analysis. Four parallel experiments were performed.

### 2.13. Statistical Analysis

All data are the mean ± SD of at least three experiments performed in duplicate (ELISA assays) or triplicate (transfection assays). Data were analyzed by unpaired one-way analysis of variance (ANOVA), or two-way analysis of variance (ANOVA) followed by Bonferroni’s post hoc test. Statistical analyses were performed using GraphPad Prism 5.02 software (GraphPad Software Inc., San Diego, CA, USA). * *p* < 0.05 was considered statistically significant. The half maximal inhibitory concentration (IC_50_) was calculated using GraphPad Prism 5.02.

## 3. Results and Discussion

### 3.1. Characterization of VVWE

The HPLC analysis of VVWE identified the following compounds: five flavonols (quercetin 3-*O*-glucoside, quercetin 3-*O*-glucuronide, kaempferol 3-*O*-glucoside, hyperoside, and rutin), two anthocyanosides (delphinidin 3-*O*-glucoside and cyanidin 3-*O*-glucoside), and caftaric acid (Table 1). Quercetin-3-*O*-glucuronide, quercetin-3-*O*-glucoside, and caftaric acid were, in order, the most abundant phenols in the extract; in particular, the quercetin-3-*O*-glucuronide value was 29.14 ± 1.92 mg/g and the quercetin-3-*O*-glucoside value was 21.68 ± 0.91 mg/g (mean ± SD).

### 3.2. VVWE Reduces IL-8 Release and Promoter Activity Induced by Pro-Inflammatory Mediators

Skin inflammatory diseases are characterized by over-expression of a multitude of pro-inflammatory mediators which impact on the cells occurring in the epidermis, mostly keratinocytes.

IL-8 is one of the main chemokines released by keratinocytes during inflammatory processes, which can in turn recruit leukocytes at the site of inflammation. In the following experiments the ability of VVWE to inhibit IL-8 secretion in HaCaT cells, an immortalized cell line widely used as a model of human keratinocytes, overcoming the potential challenge of donor variation, was tested.

To test the effect of the extract on IL-8 release induced by different inflammatory conditions, cells were challenged with the endogenous stimulus TNF-α or with pure lipopolysaccharide (LPS) from *Escherichia coli* (*E. coli*), which mimics bacterial inflammation.

The extract was able to reduce TNF-α or LPS-induced IL-8 release in a concentration dependent fashion (Figure 1A,B). IL-8 secretion was more pronounced when TNF-α was used as stimulus compared to LPS; in both cases VVWE (50 μg/mL) reduced the IL-8 secretion close to the basal level (IC_50_ 2.60 and 14.04 μg/mL, respectively, for IL-8 induced by TNF-α or LPS).

To clarify if the effect of VVWE on the IL-8 release could be due to impairment of the corresponding promoter activity, HaCaT cells were transfected by IL-8-LUC plasmid as described in the materials and method section.

The extract was able to reduce IL-8 promoter activity although the effect was less pronounced when compared to the ability to inhibit IL-8 release induced by pro-inflammatory stimuli (Figure 2A,B). The IC_50_ was 22.73 μg/mL on the TNF-α-induced IL-8 promoter activity, whereas IC_50_ was 33.98 μg/mL on the LPS-induced IL-8 promoter activity.

The effect of VVWE on IL-8 secretion induced by H_2_O_2_, as a pro-oxidant, was also evaluated. H_2_O_2_ doubled the amount of IL-8 released by HaCaT cells, but VVWE reduced the chemokine release just at the highest concentration (50 μg/mL, data not shown) thus suggesting that the effect of the extract is higher when the chemokine is released by pro-inflammatory mediators.

### 3.3. VVWE Impairs the NF-κB Pathway Acting on Transcription and Nuclear Translocation

NF-κB represents a key factor in a variety of skin inflammatory conditions including psoriasis [34], and TNF-α strongly induces activation of the NF-κB pathway. NF-κB driven transcription and nuclear translocation were assessed to better clarify the involvement of this transcription factor in the mode of action elicited by the extract. Cells were transiently transfected by NF-κB-LUC plasmid and treated as previously described. TNF-α approximately doubled the NF-κB driven transcription, while LPS showed a slight minor effect. VVWE had an inhibition trend on the NF-κB driven transcription, but the effect was statistically significant only when cells were challenged with TNF-α as stimulus (Figure 3A,B).

VVWE was also able to impair the NF-κB nuclear translocation induced by TNF-α, with a reduction of 50% at 50 μg/mL (Figure 4A). LPS showed weaker induction of translocation compared to the cytokine, whereas VVWE completely abolished LPS-induced nuclear translocation at 25 μg/mL (Figure 2D). Furthermore, VVWE (50 μg/mL) reduced the LPS-induced NF-κB nuclear translocation below the unstimulated control level by 50%.

Our groups previously demonstrated that VVWE is able to inhibit the NF-κB pathway through the impairment of the translocation from the cytoplasm into the nucleus in human gastric epithelial cells [30]. However, this is the first study reporting the effect of grapevine leaves as inhibitors of the NF-κB driven transcription and nuclear translocation induced by different pro-inflammatory stimuli.

### 3.4. VVWE Reduces IL-8 Release and NF-κB Nuclear Translocation Induced by UV-B Radiations

HaCaT cells and primary human keratinocytes display distinct keratinocyte morphology and undergo UVB-induced apoptosis [35]. UV-B induces oxidative stress in keratinocytes through the generation of reactive oxygen species (ROS) and activates several inflammatory pathways such as the mitogen-activated protein kinase (MAPK), NF-κB, and Janus kinase (JAK)/signal transduction and activation of transcription (STAT) signaling [10]. Keratinocytes are the major target of UV-B radiation and their response is predominantly regulated by the NF-κB.

VVWE reduced the UV-B induced IL-8 secretion (Figure 3A) at basal level at 100 μg/mL (IC_50_ 2.42 μg/mL) and the effect of the highest concentration paralleled the effect on the UV-B induced nuclear translocation (Figure 3B).

### 3.5. Effects of VVWE on VEGF and MMP-9 Release

VVWE was assayed on the ability to influence the release of markers widely involved in skin pathological processes including inflammatory-based conditions and wound injury. In particular, VEGF is a key regulator of the angiogenesis process whereas MMP9 is involved in the extracellular matrix remodeling. TNF-α and LPS increased the amount of VEGF approximately to 200% compared to the basal control level, and VVWE reduced the secretion significantly starting from 10 or 25 μg/mL (Figure 4A,B, respectively). H_2_O_2_ was able to induce VEGF release and the extent was comparable to that caused by TNF-α; the extract counteracted the release in a concentration-dependent fashion, with IC_50_ of 27.26 μg/mL (Figure 4C). The inhibition of MMP-9 secretion was evaluated in HaCaT cells by VVWE. Treatment for 24 h with TNF-α or LPS induced a 7.3- or 1.6-fold increase of MMP-9, respectively. VVWE (10–50 μg/mL) was not able to inhibit TNF-α-induced MMP-9 release (Figure 5A); surprisingly, VVWE showed further induction of MMP-9 released by LPS, with a three-fold at 50 μg/mL with respect to the unstimulated control (Figure 5B).

In contrast, administration of VVWE at 50 μg/mL in the absence of pro-inflammatory stimuli induced an increase of MMP-9 secretion of 1.4-fold compared to control, suggesting a synergistic effect with LPS. VVWE (200 μg/mL), without other pro-inflammatory stimuli, induced the release of MMP-9 in a concentration-dependent manner up to 6.2-fold with respect to control (data not shown).

### 3.6. Franz Diffusion Cell Method

In order to evaluate the ability of the extract components to permeate the epidermis and therefore their possible bioavailability, VVWE was assayed on Franz diffusion cells over a period of 48h. Among all compounds identified in the extract only a limited part of them pass through the epidermis (Table 1) during the permeation process. In very few cases and in small amounts these compounds were detected before 48 h in the receiver phase.

Starting from 150 mg of VVWE, the most abundant compounds able to cross the skin barrier were in the following order: quercetin 3-*O*-glucuronide (51.66 µg/g), caftaric acid (47.59 µg/g), and quercetin 3-*O*-glucoside (8.74 µg/g) (Table 1). The analysis of the compounds retained into the skin’s portion revealed that the most abundant were quercetin 3-*O*-glucuronide (28.08 µg/g) > quercetin 3-*O*-glucoside (13.58 µg/g) > caftaric acid (10.20 µg/g). Being an aqueous extract, VVWE contains hydrophilic compounds that are less suitable for skin permeation. Ethanol, as a permeation enhancer, was added to improve solubility and to disorder skin lipids [36]. Antioxidants detected in the receiver phase and into the epidermis have log*p* values below 1.15 (caftaric acid) and 1.21 (kaempferol 3-*O*-glucoside; from predicted properties, SciFinder). In the case of 3-*O*-glucoside derivatives, the predicted properties were not found other than for kaempferol and quercetin. Hyperoside and quercetin 3-*O*-glucoside are highly hydrophilic (log*p* value < 0). Detection of the latter and quercetin 3-*O*-glucuronide (log*p* = 0.62) is probably due to their abundance in the donor phase. To further improve these results, a proper vehicle should be tested as already reported in case of quercetin [37].

### 3.7. Effects of Pure Compounds on IL-8 Secretion

The most abundant compounds of VVWE that were able to cross the skin barrier (caftaric acid, quercetin-3-*O*-glucoside, and quercetin-3-*O*-glucuronide) were tested separately with respect to their ability to reduce IL-8 secretion in HaCaT cells. Caftaric acid, quercetin 3-*O*-glucoside, and quercetin 3-*O*-glucuronide (0.1–100 μM) failed to reduce IL-8 release, induced by TNF-α, up to the maximum concentration tested (100 μM), thus suggesting that the effect of the extract could be due to synergistic effects occurring among the constituents or to the presence of other bioactive compounds still unknown.

## 4. Conclusions

This is the first study showing grapevine leaves as inhibitors of the NF-κB pathway at the cutaneous level. VVWE reduced two typical markers of psoriatic lesions, IL-8 and VEGF. The effect of the extract was higher when pro-inflammatory stimuli were used; however it also showed anti-oxidant mechanisms of action against H_2_O_2_ and UVB radiation. In parallel, VVWE did not inhibit MMP-9 release, potentially promoting tissue remodeling while reducing other inflammatory markers. Analytical studies showed that diffusion of polar compounds through the skin layer is markedly reduced, but still able to reach potential bioactive concentrations. Unfortunately, the evaluation of individual compounds did not identify bioactive components, and further studies are required. Taken together, our results seem to suggest the possible use of grapevine leaves as anti-inflammatory agents for skin inflammatory conditions.

## Figures and Tables

**Figure 1 antioxidants-08-00134-f001:**
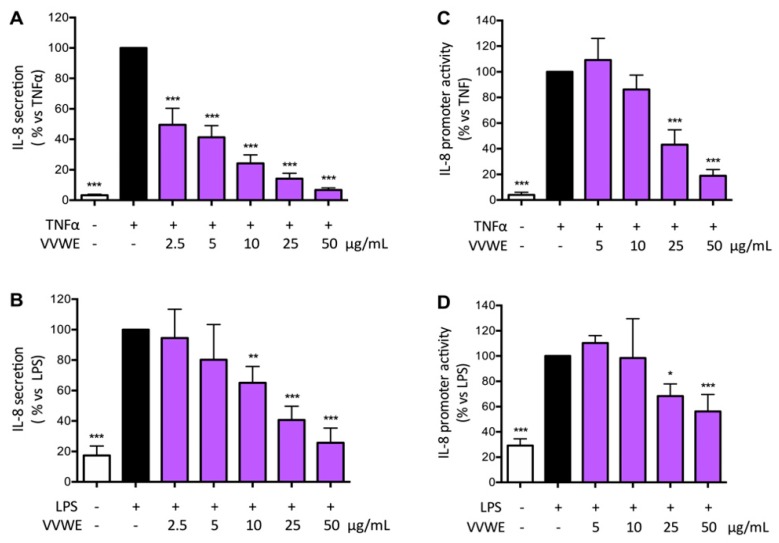
Effect of water extract from *Vitis vinifera* leaves (VVWE) on the tumor necrosis factor-α (TNF-α)-induced (**A**) or lipopolysaccharide (LPS)-induced (**B**) interleukin-8 (IL-8) secretion. HaCaT cells were treated for 6 h with TNF-α (10 ng/mL) or LPS (5 μg/mL) and VVWE (2.5–50 μg/mL). IL-8 secretion was evaluated by ELISA assay. Basal (without pro-inflammatory stimuli) and control (with TNF-α/LPS) levels of IL-8 were 17.4, 560.1 and 108.7 pg/mL, respectively. The effect of VVWE was evaluated on IL-8 promoter activity induced by TNF-α (**C**) and LPS (**D**). HaCaT cells were treated for 6 h with TNF-α (10 ng/mL) or LPS (5 μg/mL) and VVWE (5–50 μg/mL). IL-8 promoter activity was measured in transfected HaCaT cells by the luciferase assay. The graphs show the means ± SD of at least three experiments performed in triplicate. Statistical analysis: one-way ANOVA, followed by Bonferroni’s post hoc test. * *p* < 0.05, ** *p* < 0.01, *** *p* < 0.001 vs. TNF-α alone. Epigallocatechin-3-gallate (EGCG) (20 μM) was used as the reference inhibitor of TNF-α or LPS-induced IL-8 secretion (62.1% and 84.9% inhibition, respectively) and IL-8 promoter activity (35.9% and 33.3% inhibition, respectively); the effect of the reference inhibitor is in agreement with that reported in the scientific literature.

**Figure 2 antioxidants-08-00134-f002:**
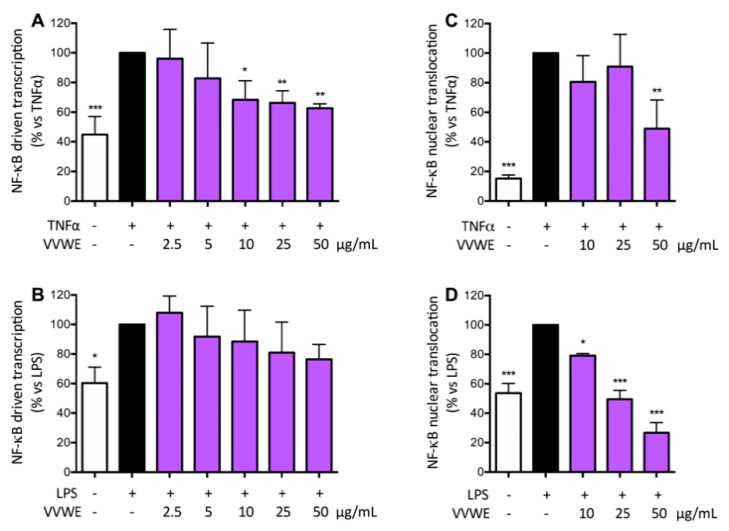
Effect of VVWE on the nuclear factor-κB (NF-κB)-driven transcription induced by TNF-α (**A**) or LPS (**B**). HaCaT cells were treated 6 h with TNF-α (10 ng/mL) or LPS (5 μg/mL) and VVWE. NF-κB nuclear translocation was evaluated in HaCaT cells treated for 1 h with TNF-α (**C**) or LPS (**D**) and VVWE (10–50 μg/mL). The graphs show the means ± SD of at least three experiments performed in triplicate and duplicate, for transcription and translocation respectively. Statistical analysis: one-way ANOVA, followed by Bonferroni’s post hoc test. * *p* < 0.05, ** *p* < 0.01, *** *p* < 0.001 vs. TNF-α or LPS alone. Here, 20 μM EGCG were used as the reference inhibitor of TNF-α or LPS-induced NF-κB-driven transcription (59.9 and 29.9% inhibition, respectively) and NF-κB nuclear translocation (62.4% and 53.2% inhibition, respectively).

**Figure 3 antioxidants-08-00134-f003:**
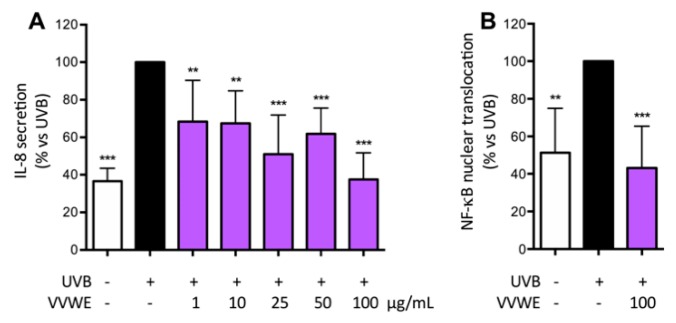
Effect of VVWE on IL-8 secretion (**A**) or NF-κB nuclear translocation (**B**) induced by ultraviolet B (UVB) irradiation. HaCaT cells were treated for 50 s with UVB (40 mJ/cm^2^) and followed by VVWE for 9 h (IL-8 secretion) or 1 h (NF-κB nuclear translocation). The graphs show the means ± SD of at least three experiments performed in triplicate and duplicate, for IL-8 secretion and translocation, respectively. Statistical analysis: one-way ANOVA, followed by Bonferroni’s post hoc test. ** *p* < 0.01, *** *p* < 0.001 vs. UVB alone. Here, 20 μM EGCG were used as the reference inhibitor for UVB-induced IL-8 secretion (61.7% inhibition) and NF-κB nuclear translocation (39.1% inhibition).

**Figure 4 antioxidants-08-00134-f004:**
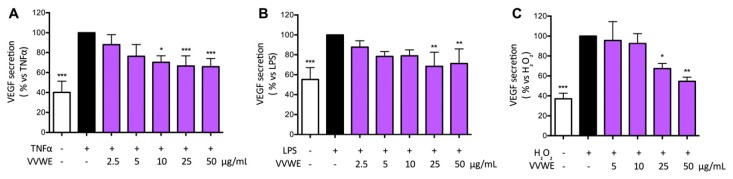
Effect of VVWE on vascular endothelial growth factor (VEGF) secretion induced by TNF-α (**A**), LPS (**B**), and H_2_O_2_ (**C**). HaCaT cells were treated for 24 h with TNF-α (10 ng/mL), H_2_O_2_ (100 μM) or 30 h with LPS (5 μg/mL) and VVWE (2.5–50 μg/mL). Secreted VEGF was evaluated by the ELISA assay. The graphs show the means ± SD of at least three experiments performed in triplicate. Statistical analysis: one-way ANOVA, followed by Bonferroni’s post hoc test. * *p* < 0.05, ** *p* < 0.01, *** *p* < 0.001 vs. TNF-α alone. Here, 20 μM EGCG were used as the reference inhibitor of TNF-α, LPS, or H_2_O_2_-induced VEGF secretion (78.7%, 100%, and 100% inhibition, respectively).

**Figure 5 antioxidants-08-00134-f005:**
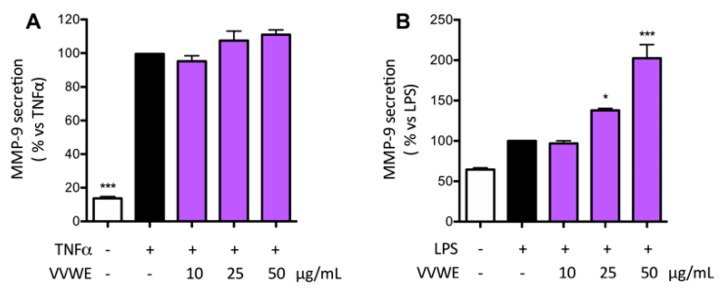
Effect of VVWE on MMP-9 secretion induced by TNF-α (**A**) or LPS (**B**). HaCaT cells were treated for 24 h with TNF-α (10 ng/mL) or 30 h with LPS (5 μg/mL) and VVWE (10–50 μg/mL). Matrix metalloproteinase-9 (MMP-9) release was evaluated by the ELISA assay. The graphs show the means ± SD of at least three experiments performed in triplicate. Statistical analysis: one-way ANOVA, followed by Bonferroni’s post hoc test. * *p* < 0.05, *** *p* < 0.001 vs. TNF-α alone. EGCG (20 μM) was used as the reference compound of TNF-α or LPS-induced MMP-9 secretion (38.0% and no inhibition, respectively).

**Table 1 antioxidants-08-00134-t001:** Data are expressed as mg(^a^) or µg(^b^) of pure compound per g water extract from *Vitis vinifera* leaves (VVWE) (mean ± SD). Recovery percentage (% recovery) was calculated on the availability in weight of pure compounds after permeation of 150 mg of VVWE in four replicates.

VVWE Composition: Identified Compounds	Donor Solution Content mg/g ^a^ ± SD	Permeated Amount µg/g ^b^ ± SD (% Recovery)	Retained Amount in the Epidermidis µg/g ^b^ ± SD (% Recovery)
Caftaric acid	9.99 ± 0.35	47.59 ± 27.64 (0.476%)	10.20 ± 9.00 (0.102%)
Rutin	1.31 ± 0.05	0.30 ± 0.35 (0.023%)	0.80 ± 0.91 (0.061%)
Hyperoside	2.30 ± 0.17	0.47 ± 0.68 (0.020%)	1.34 ± 1.43 (0.058%)
Quercetin 3-*O*-glucoside	21.68 ± 0.91	8.74 ± 9.50 (0.040%)	13.58 ± 15.88 (0.063%)
Quercetin 3-*O*-glucuronide	29.14 ± 1.92	51.66 ± 46.10 (0.177%)	28.08 ± 30.33 (0.096%)
Kaempferol 3-*O*-glucoside	3.77 ± 0.06	2.09 ± 1.70 (0.055%)	3.53 ± 3.90 (0.094%)
Delphinidin 3-*O*-glucoside	0.95 ± 0.03	0.48 ± 0.68 (0.050%)	0.04 ± 0.08 (0.004%)
Cyanidin 3-*O*-glucoside	2.29 ± 0.04	1.63 ± 2.08 (0.071%)	0.30 ± 0.60 (0.013%)
Petunidin 3-*O*-glucoside	0.66 ± 0.05	0.38 ± 0.53 (0.058%)	0.10 ± 0.20 (0.015%)
Peonidin 3-*O*-glucoside	1.91 ± 0.06	2.17 ± 2.72 (0.114%)	0.58 ± 1.17 (0.030%)
Malvidin 3-*O*-glucoside	1.27 ± 0.07	0.92 ± 1.22 (0.072%)	0.29 ± 0.58 (0.023%)

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
