# Peer review of "Vitis vinifera L. Leaf Extract Inhibits In Vitro Mediators of Inflammation and Oxidative Stress Involved in Inflammatory-Based Skin Diseases"

_antioxidants, 2019, doi:10.3390/antiox8050134_

Round 1
Reviewer 1 Report
This paper is very well written and presents some interesting and useful information on the ant-inflammatory properties of vine leaf extracts. The methods used are sound and the results are presented in a clear fashion. I recommend publication subject to some very minor amendments. See details below.
Lines 61-62: “Furthermore, TNFα regulates extracellular matrix remodeling through matrix metalloproteinases (MMPs) production in keratinocytes [17] including MMP-9 [18]; MMPs are deeply involved in cell migration, tissue remodeling, vasodilatation and angiogenesis.”, to
“Furthermore, TNFα regulates extracellular matrix remodeling through matrix metalloproteinases (MMPs) production in keratinocytes [17] including MMP-9 [18]. MMPs are deeply involved in cell migration, tissue remodeling, vasodilatation and angiogenesis.” (i.e. replace “;” with “.”).
Line 70: Replace “psoriasis” with “psoriasis, ”.
Line 73: Replace “...conditions [21]; however,...” with “...conditions [21]. However,...” (i.e. replace “;” with “.”).
Line 75: Change “originated” to “originating”.
Line 79: Change “...and anthocyanins; moreover, grapevine leaves...” to “...and anthocyanins; moreover, grapevine leaves...”.
Line 84: Change “...use; however...” to “...use. However...”.
Line 143: Change “104” to “104”.
Line 153: “...translocation HaCaT...” to “...translocation, HaCaT...”.
Line 166: Change “cm2” to “cm2”.
Line 224-226: Change “In the following experiments the ability of VVWE to inhibit IL-8 secretion in HaCaT cells, which is immortalized cell line widely used as a model of human keratinocytes, overcoming the potential challenge of donor variation.”, to
“In the following experiments the ability of VVWE to inhibit IL-8 secretion in HaCaT cells, which is immortalized cell line widely used as a model of human keratinocytes, overcoming the potential challenge of donor variation, was tested.”
Line 250: Change “if” to “when” (end of line).
Line 252: Change “...whereas was...” to “...whereas IC50 was...”.
Author Response
Reviewer 1
Comments to authors
Point 1:This paper is very well written and presents some interesting and useful information on the ant-inflammatory properties of vine leaf extracts. The methods used are sound and the results are presented in a clear fashion. I recommend publication subject to some very minor amendments. See details below.
Response 1: The authors thank the reviewer 1 for positive comments.
Point 2: Lines 61-62: “Furthermore, TNFα regulates extracellular matrix remodeling through matrix metalloproteinases (MMPs) production in keratinocytes [17] including MMP-9 [18]; MMPs are deeply involved in cell migration, tissue remodeling, vasodilatation and angiogenesis.”, to “Furthermore, TNFα regulates extracellular matrix remodeling through matrix metalloproteinases (MMPs) production in keratinocytes [17] including MMP-9 [18]. MMPs are deeply involved in cell migration, tissue remodeling, vasodilatation and angiogenesis.” (i.e. replace “;” with “.”).
Response 2: text was changed accordingly.
Point 3: Line 70: Replace “psoriasis” with “psoriasis, ”.
Response 3: text was changed accordingly.
Point 4: Line 73: Replace “...conditions [21]; however,...” with “...conditions [21]. However,...” (i.e. replace “;” with “.”).
Response 4: text was changed according to reviewer suggestion.
Point 5: Line 75: Change “originated” to “originating”.
Response 5: text was changed according to reviewer correction.
Point 6:Line 79: Change “...and anthocyanins; moreover, grapevine leaves...” to “...and anthocyanins; moreover, grapevine leaves...”.
Response 6: text was changed according to reviewer correction.
Point 7:Line 84: Change “...use; however...” to “...use. However...”.
Response 7: text was changed according to reviewer correction.
Point 8:Line 143: Change “104” to “104”.
Response 8: text was changed according to reviewer correction.
Point 9:Line 153: “...translocation HaCaT...” to “...translocation, HaCaT...”.
Response 9: text was changed according to reviewer correction.
Point 10:Line 166: Change “cm2” to “cm2”.
Response 10: text was changed according to reviewer correction.
Point 11:Line 224-226: Change “In the following experiments the ability of VVWE to inhibit IL-8 secretion in HaCaT cells, which is immortalized cell line widely used as a model of human keratinocytes, overcoming the potential challenge of donor variation.”, to “In the following experiments the ability of VVWE to inhibit IL-8 secretion in HaCaT cells, which is immortalized cell line widely used as a model of human keratinocytes, overcoming the potential challenge of donor variation, was tested.”
Response 11: text was changed according to reviewer correction.
Point 12:Line 250: Change “if” to “when” (end of line). Line 252: Change “...whereas was...” to “...whereas IC50 was...”.
Response 12: text was changed according to reviewer correction.
Reviewer 2 Report
Sangiovanni et al. tested the ability of a water extract from grapevine leaves to inhibit inflammatory conditions in human keratinocytes, challenged with proinflammatory (TNFα or LPS) or prooxidant (UVB or H2O2) mediators. The obtained results are interesting and promising for the application of the extract against inflammatory-based skin diseases.
The paper is suitable for publication in Antioxidants after minor revisions.
1) In the Introduction (page 2, lines 72-74) , the authors should stress the relevant antioxidant and inflammatory activities of natural extracts obtained from plant materials including several references (see for example: Food Chem. 2019, 289, 56-64. doi: 10.1016/j.foodchem.2019.02.127; J. Ethnopharmacol. 2019, 233, 41-46. doi: 10.1016/j.jep.2018.12.044; Nutrients 2019, 11, 548-562. doi: doi:10.3390/nu11030548; J.Cosm.Dermatol. 2018; doi.org/10.1111/jocd.12842 …………) and then specify the lack of literature on the topical use in psoriasis.
2) In Results and Discussion the characterization of the extract should be the 3.1 paragraph.
3) Please include the title of ref.14.
Author Response
Reviewer 2
Comments to authors
Point 1:Sangiovanni et al. tested the ability of a water extract from grapevine leaves to inhibit inflammatory conditions in human keratinocytes, challenged with proinflammatory (TNFαor LPS) or prooxidant (UVB or H2O2) mediators. The obtained results are interesting and promising for the application of the extract against inflammatory-based skin diseases.
The paper is suitable for publication in Antioxidants after minor revisions.
Response 1:The authors thank the reviewer 2 for appreciation of the work and for useful comments.
Point 2: In the Introduction (page 2, lines 72-74) , the authors should stress the relevant antioxidant and inflammatory activities of natural extracts obtained from plant materials including several references (see for example: Food Chem. 2019, 289, 56-64. doi: 10.1016/j.foodchem.2019.02.127; J. Ethnopharmacol. 2019, 233, 41-46. doi: 10.1016/j.jep.2018.12.044; Nutrients 2019, 11, 548-562. doi: doi:10.3390/nu11030548; J.Cosm.Dermatol. 2018; doi.org/10.1111/jocd.12842 …………) and then specify the lack of literature on the topical use in psoriasis.
Response 2: In the Introduction section the authors have added a description of natural compounds showing relevant antioxidant activities making examples and citing the literature suggested by the reviewer. We have also specified the lack of literature on the topical use in psoriasis.
Point 3:In Results and Discussion the characterization of the extract should be the 3.1 paragraph.
Response 3: Organization of paragraphs has been changed according to reviewer suggestion.
Point 4:Please include the title of ref.14.
Response 4: the title was included in the previous version of the manuscript and is “psoriasis”.